

# Sideritis scardica extracts inhibit aggregation and toxicity of amyloid-β in Caenorhabditis elegans used as a model for Alzheimer's disease

Felix Heiner[1], Björn Feistel[2] and Michael Wink[1]

[1] Institute of Pharmacy and Molecular Biotechnology, Heidelberg University, Heidelberg, Germany
[2] Finzelberg GmbH & Co. KG, Andernach, Germany

## ABSTRACT

**Background.** Beyond its traditional uses in the Balkan area, *Sideritis scardica* (known as Greek mountain tea, Lamiaceae) is currently extensively investigated for its pharmacological activity in the central nervous system. Antidepressant, psychostimulating, cognition-enhancing and neuroprotective properties have been described. In this study, we tested hydroalcoholic extracts of *S. scardica* for their potential to counteract amyloid-β toxicity and aggregation, which plays a crucial role in the pathogenesis of Alzheimer's disease.

**Methods.** For this purpose, we have chosen the nematode *Caenorhabditis elegans*, which is used as a model organism for neurodegenerative diseases. The concentration of different polyphenols in extracts prepared from water, 20, 40, 50, and 70% ethanol was analysed by HPLC. Additionally, polar and unpolar fractions were prepared from the 40% ethanolic extract and phytochemically analysed.

**Results.** Essentially, the contents of all measured constituents increased with the lipophilicity of the extraction solvents. Treatment of transgenic *C. elegans* strains expressing amyloid-β with the extracts resulted in a reduced number of peptide aggregates in the head region of the worms and alleviated toxicity of amyloid-β, observable through the degree of paralysed animals. The mid-polar extracts (40 and 50% ethanol) turned out be the most active, decreasing the plaque number by 21% and delaying the amyloid-β-induced paralysis by up to 3.5 h. The more lipophilic extract fractions exhibited higher activity than the hydrophilic ones.

**Discussion.** *Sideritis scardica* extracts demonstrated pharmacological activity against characteristics of Alzheimer's disease also in *C. elegans*, supporting current efforts to assess its potential for the treatment of cognitive decline. The active principle as well as the mode of action needs to be investigated in more detail.

## INTRODUCTION

Alzheimer's disease (AD) is the most common type of dementia and also the most common neurodegenerative disorder in general. As nowadays more people reach a high age than

Corresponding author
Felix Heiner, felixheiner@gmail.com

in the past and a cure is still missing, AD is a rising concern for modern civilizations. According to the World Alzheimer Report 2015, about 47 million people suffered from dementia in 2015, and numbers may double every 20 years (*Prince et al., 2015*). A variety of possible causes for AD are being discussed, of which amyloid-β peptides (Aβ) still play a key role in Alzheimer research and which could be targeted by drugs therapeutically or preventively (*Hardy & Selkoe, 2002*). Aβ is derived from the amyloid precursor protein (APP) by cleavage through β- and γ-secretases (*Selkoe, 1997*). The monomers aggregate to oligomers, to polymers and finally to senile plaques, which are abundant in the brain of patients suffering from AD (*Lansbury, 1999*). The traditional formulation of the amyloid hypothesis blamed those mature aggregates for neurodegeneration, but the smaller oligomers were discovered to be the most neurotoxic Aβ species (*Lambert et al., 1998*; *Walsh & Selkoe, 2004*).

*Sideritis scardica* Griseb. (Lamiaceae) is a perennial shrub endemic to the Balkan peninsula, with Bulgaria as its main habitat. Depending on the area, it is commonly known as Greek mountain tea, Shepherd's tea, Ironwort, Mursalski tea, Pirinski tea, or Caj Mali. A broad range of traditional uses of *S. scardica* are known, including the treatment of bronchitis, asthma, sore throat, the prevention of anemia, and the use as tonic or poultice (*Todorova & Trendafilova, 2014*). Concerning the traditional use against cough associated with common cold and gastrointestinal discomfort, a *HMPC (2016)* monograph is available. The plant is rich in polyphenols, such as flavonoids, hydroxycinnamic acid derivatives, and phenylethanoid glycosides (*Evstatieva, 2002*; *Petreska et al., 2011*). Pharmacological activities like antimicrobial, gastroprotective and anti-inflammatory activity are mostly accredited to this class of secondary metabolites (*Tadic et al., 2007*; *Tadic et al., 2012a*; *Tadic et al., 2012b*). Recently the effects of *S. scardica* extracts on the central nervous system were addressed in a number of studies. Hydroalcoholic extracts were able to inhibit the reuptake of the monoamine neurotransmitters noradrenaline, dopamine and serotonin *in vitro* (*Feistel & Appel, 2013*; *Knörle, 2012*). Furthermore, they showed antidepressant and psychostimulating effects, as well as a modulation of AMPA-dependent neurotransmission in rats (*Dimpfel, 2013*; *Dimpfel, Schombert & Feistel, 2016a*). In mice, cognitive enhancement and Aβ-counteracting effects were observed (*Hofrichter et al., 2016*). Also, clinical studies have already been performed. *S. scardica* extracts were able to improve the mental performance of healthy subjects under stress conditions and of subjects suffering from mild cognitive impairment (MCI), which is a precursor of AD (*Behrendt et al., 2016*; *Dimpfel, Schombert & Biller, 2016b*). A double-blind, randomized, and placebo-controlled clinical trial currently demonstrates a significant effect of a combination of *S. scardica* and *Bacopa monnieri* extract (memoLoges®) on the mental performance of subjects suffering from MCI (*Dimpfel et al., 2016c*).

To further investigate the influence of hydroalcoholic *S. scardica* extracts on neurodegenerative diseases and especially on Aβ toxicity and aggregation, we have chosen *Caenorhabditis elegans* as a model organism (*Link, 2006*). In transgenic strains expressing human Aβ (1–42), *in vivo* effects can be observed, that, unlike *in vitro* studies, also consider bioavailability and other biological influences on a multicellular organism. In the present study, we also tried to figure out the influence of extraction solvents on the content of
polyphenolic compounds and pharmacological activity, if a dose-dependency exists, and which extract fractions are potent in order to explore the active principle.

## MATERIALS AND METHODS

### Plant material

The drug Sideritidis scardicae herba from cultivation in Bulgaria was obtained from Finzelberg GmbH & Co. KG, Andernach, Germany (Item 2232000; Batch 10018839). Voucher specimens are deposited at the Department of Biology, Institute of Pharmacy and Molecular Biotechnology, Heidelberg University, Germany (registration number P8562) and at the Department of Pharmacognosy and Natural Products Chemistry, Faculty of Pharmacy, University of Athens, Greece (Specimen-No. PAS101), where the plant material was identified and specified. Five crude extracts with water, ethanol 20%, 40%, 50%, and 70% (V/V) were prepared by exhaustive extraction with twofold moved maceration. After filtering and uperisation (3 s at 120 °C) they were dried under vacuum. The 40% ethanolic extract was additionally fractionated through liquid–liquid extraction (aqueous and butanolic phase), reprecipitation in 70% ethanol (V/V) (supernatant and precipitate) and solid–liquid separation with an Amberlite® XAD7HP (Sigma-Aldrich, St. Louis, USA) adsorber resin (aqueous and ethanolic phase). For the latter one, an aqueous solution of the primary extract was applied onto the column and the compounds were eluted with water and, subsequently, with increasing concentrations of ethanol. All test substances were stored at 4 °C.

### Phytochemical analysis

The extracts and fractions were analysed for total polyphenols with a Folin-Ciocalteu UV method following chapter 2.8.14. of the *European Pharmacopoeia (2017a)* and for specific polyphenolic compounds (flavonoids, acteoside, caffeoylquinic acids) with a HPLC method. For this purpose, a Luna® C18/2 column (Phenomenex, Torrance, CA, USA; 250 mm length, 4.6 mm inner diameter, 5 $\mu$m particle size) was used at a temperature of 40 °C in a Shimadzu LC10 HPLC system. 10 $\mu$L of about 5 mg/mL sample were injected. The mobile phase was composed of water + 0.1% $H_3PO_4$ ($H_2O$) and acetonitrile + 0.1% $H_3PO_4$ (ACN) with the following gradient: From 95% $H_2O$/5% ACN (0 min) to 50% $H_2O$/50% ACN in 41 min; 100% ACN from 45 to 50 min to 95% $H_2O$/5% ACN until 52 min; 65 min in total. The compounds were detected by DAD at 330 nm and calculated through scutellarin, chlorogenic acid, and acteoside (Phytolab, Vestenbergsgreuth, Germany) as external standards (Fig. 1). Additionally, thin layer chromatography (TLC) was conducted to highlight differences of the fractions. As the stationary phase, silica gel 60 $F_{254}$ was used. The plate was cleaned and activated with ethyl acetate/methanol 50:50 (V/V) and dried at 105 °C for 30 min. 10 $\mu$L of preparations from 1 g *S. scardica* extract and 10 mL ethanol 50% (10 min at 65 °C, filtered) were applied and separated within 15 cm using dichloroethane/acetic acid/methanol/water 50:25:15:10 (V/V/V/V) as mobile phase. After drying, anisaldehyde solution R (*European Pharmacopoeia, 2017b*) was sprayed on the plate, which was dried again for 3 min at 120 °C.

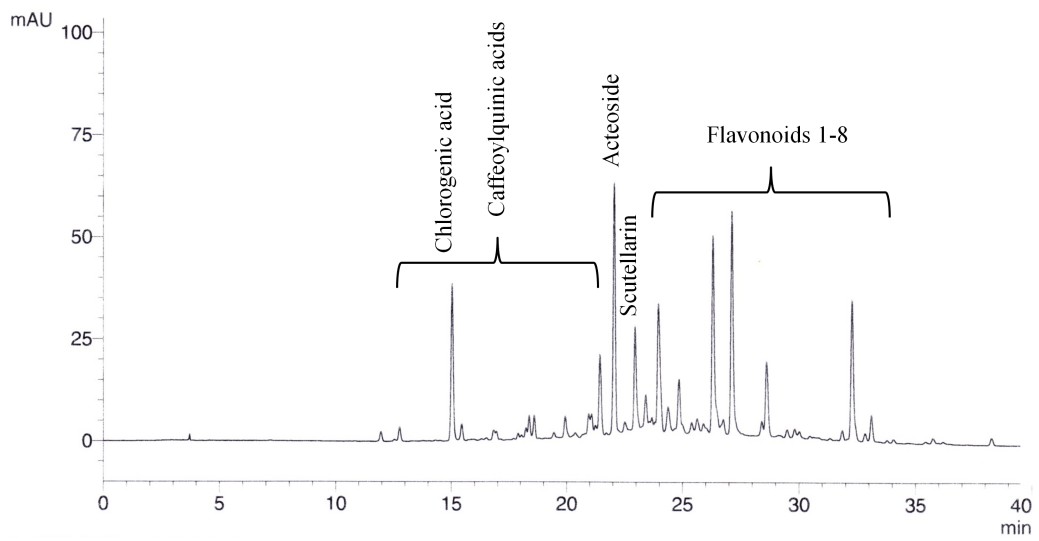

**Figure 1 Exemplary chromatogram of 20% ethanolic *S. scardica* extract.** Caffeoylquinic acids (calculated as chlorogenic acid), acteoside, and flavonoids (calculated as scutellarin) could be quantified by HPLC with UV detection at 330 nm.

### *C. elegans* strains and culture conditions

Transgenic *C. elegans* strain CL2006 (genotype *dvIs2 [pCL12(unc-54/human Abeta peptide 1–42 minigene) + pRF4]*) constitutively expresses human Aβ (1–42) in its muscle cells. In strain CL4176 (genotype *smg-1(cc546) I; dvIs27 [myo-3p::A-Beta (1–42)::let-851 3′ UTR) + rol-6(su1006)] X*) the Aβ expression is temperature inducible through mutation of *smg-1*. Strain CL802 (genotype *smg-1(cc546) I; rol-6(su1006) II*) also possesses the mutated *smg-1* gene but is not able to express Aβ, representing a suitable control for CL4176. Additionally, both strains used for paralysis assay contain a roller marker for visual discrimination of phenotypes. All strains were obtained from the Caenorhabditis Genetics Center. The worms were cultured on nematode growth medium (NGM) with *E. coli* OP50 as a food source at 20 °C (strain CL2006) or 16 °C (CL4176, CL802). To start with age-synchronized worms, a hypochlorite treatment of gravid adults for 8 min, which isolates the eggs, was performed before every assay (1% NaOCl, 0.5 M NaOH; Sigma-Aldrich, St. Louis, MI, USA).

### Quantification of β-amyloid aggregates

Isolated eggs of strain CL2006 were incubated in S-medium containing about $10^9$ *E. coli* OP50/mL for 48 h at 16 °C. The hatched worms were then transferred to NGM plates containing the desired concentration of the test substances and *E. coli* OP50. 100 μg/mL EGCG (Sigma-Aldrich, St. Louis, MI, USA) from green tea served as a positive control. After 96 h of incubation at 16 °C the worms were fixed and Aβ aggregates were stained with 0.0125% thioflavin S (Sigma-Aldrich, St. Louis, MI, USA) in 50% ethanol as described before (*Fay et al., 1998*). The Aβ plaques in the head region of 20–25 worms per treatment were counted using a Keyence BZ-9000 fluorescence microscope with a GFP filter (excitation wavelength 480 nm, emission wavelength 510 nm).

### Paralysis assay (Aβ toxicity)

The assay was performed as described before (*Dostal & Link, 2010*). In brief, the treated worms were kept at 16 °C for 48 h, then the temperature was upshifted to 25 °C to induce the expression of Aβ. On the next day, scoring was conducted at least every 2 h for at least 12 h. The worms were counted as paralysed if they failed to respond to several touches with a platinum wire.

### Statistical analysis

All results are expressed as the mean ± S.E.M of at least three independently repeated experiments. The median paralysis times ($PT_{50}$) were obtained with a Kaplan–Meier survival analysis. One-way ANOVA with Bonferroni *post-hoc* correction/independent two-sample Student's *t*-tests (equal variance) were carried out to analyze statistical differences (as appropriate).

## RESULTS

The phytochemical analysis revealed that basically the content of all tested plant compounds increased with decreasing polarity of the extraction solvent (Table 1). Nevertheless, compared to extraction solvents of stronger lipophilicity, the amount of total phenols in the 40% ethanolic extract was surprisingly high which may be based on the lower drug-extract ratio. From water to 70% ethanol the content of acteoside was enriched almost 13-fold in concentration. The more lipophilic fractions of the 40% ethanolic primary extract (Liq-Liq BuOH, Reprecip. supernat, Resin EtOH) showed higher concentrations of total polyphenols, acteoside and caffeoylquinic acids than the polar ones (Liq-Liq $H_2O$, Reprecip. precip., Resin $H_2O$) and also compared to the primary extract (Fig. 2). The butanolic fraction of the liquid–liquid extraction contained especially high yields of the analyzed polyphenolic compounds. It contained concentrations of acteoside and flavonoids that were approximately three times higher than the primary extract. Taking the distribution of mass into consideration, acteoside especially seemed to selectively accumulate in lipophilic solvents used for fractionation. The amount of flavanoids in the fractions obtained from reprecipitation in 70% ethanol constituted the only case of a lower concentration in the unpolar fraction (supernatant) compared to the polar one (precipitate) and to the primary extract. However, the TLC consistently displays the generally higher content of polyphenolic constituents in the lipophilic fractions (flavonoids Rf 0.3–0.5 and tannins/hydroxycinnamic acids Rf 0.7–0.8) in comparison to those with stronger polarity and to the original extract (Fig. 3).

The transgenic *C. elegans* strain CL2006 constitutively expresses human Aβ (1–42) (*Link, 1995*). These peptides form aggregates, which were stained with thioflavin S for quantification; Fig. 4 shows the visualized plaques in the head region of the worms (Fig. 4A). In a concentration of 600 μg/mL all *S. scardica* extracts significantly reduced the number of Aβ aggregates (Fig. 4B). The extract made of 20% ethanol clearly showed a concentration-dependent activity (Fig. 4C), whereas the treatment with 50% ethanolic extract seemed to lose effectiveness when the concentration was raised from 400 to 600 μg/mL. Worms that were treated with 400 μg/mL of lipophilic fractions prepared from

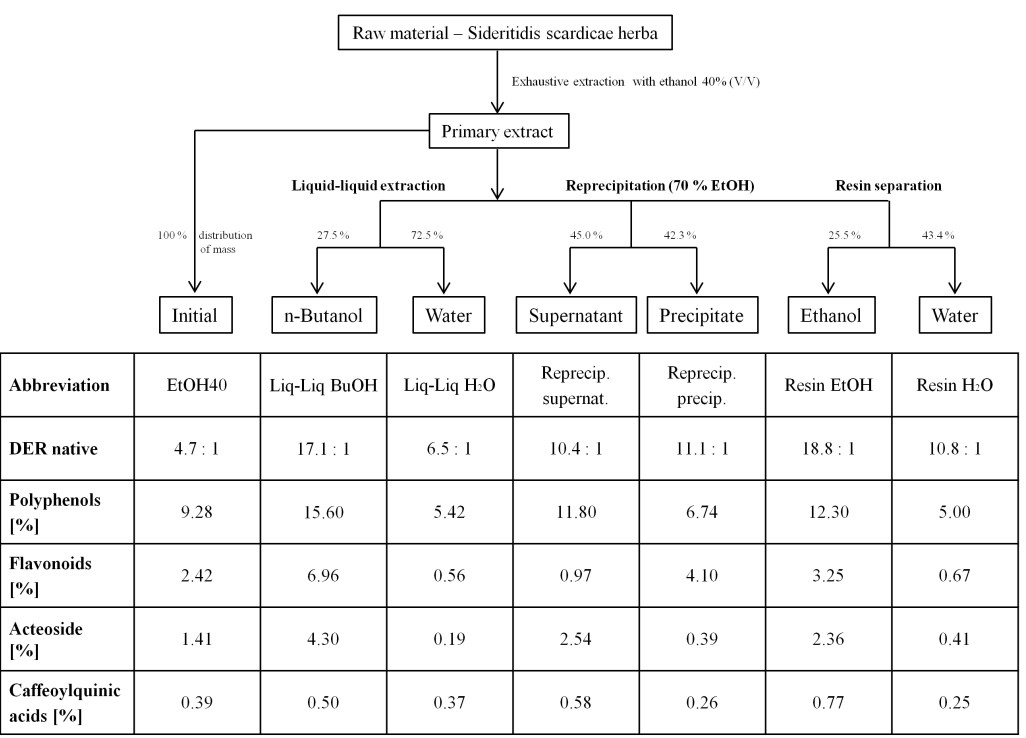

**Figure 2** **Flowchart and phytochemical analysis of the fractions prepared from 40% ethanolic *S. scardica* extract.** Percentage distribution of mass is given for each pair of fractionation type in the flowchart above. Occurring differences to 100% are loss of preparative handling.

**Table 1** **Phytochemical analysis of the *S. scardica* extracts prepared with different solvents.** Total content of polyphenolic compounds was measured with an unspecific Folin-Ciocalteu method. Single groups of polyphenols were analysed by HPLC with UV detection.

| Abbreviation | H₂O | EtOH20 | EtOH40 | EtOH50 | EtOH70 |
|---|---|---|---|---|---|
| **Extraction solvent** | Water | 20% ethanol (V/V) | 40% ethanol (V/V) | 50% ethanol (V/V) | 70% ethanol (V/V) |
| **DER native** | 5.8:1 | 7.2:1 | 4.7:1 | 5.7:1 | 5.7:1 |
| **Polyphenols [%]** | 5.07 | 6.25 | 9.28 | 6.23 | 7.37 |
| **Flavonoids [%]** | 0.59 | 1.18 | 2.42 | 2.03 | 2.82 |
| **Acteoside [%]** | 0.12 | 0.41 | 1.41 | 0.94 | 1.54 |
| **Caffeoylquinic acids [%]** | 0.24 | 0.47 | 0.39 | 0.38 | 0.49 |

the 40% ethanolic extract showed similar plaque numbers to the actual primary extract in the same concentration (Fig. 4D). In contrast, the hydrophilic fractions showed weak or no significant activity (water phase of resin separation: 10.9 ± 0.9% reduction). Taken together, the lowest numbers of Aβ plaques were seen in worms treated with 1,000 μg/mL of the 20%, 400 and 600 μg/mL of the 40%, and 400 μg/mL of the 50% ethanolic extract (20.3 ± 1.4–22.4 ± 0.4% reduction), which was slightly better than the positive control EGCG (19.5 ± 2.1%) studied previously in our laboratory (*Abbas & Wink, 2010*).

The temperature-inducible expression of human Aβ (1–42) makes worms of strain CL4176 paralyse over time, which is an outcome of Aβ toxicity (*Dostal & Link, 2010*). The

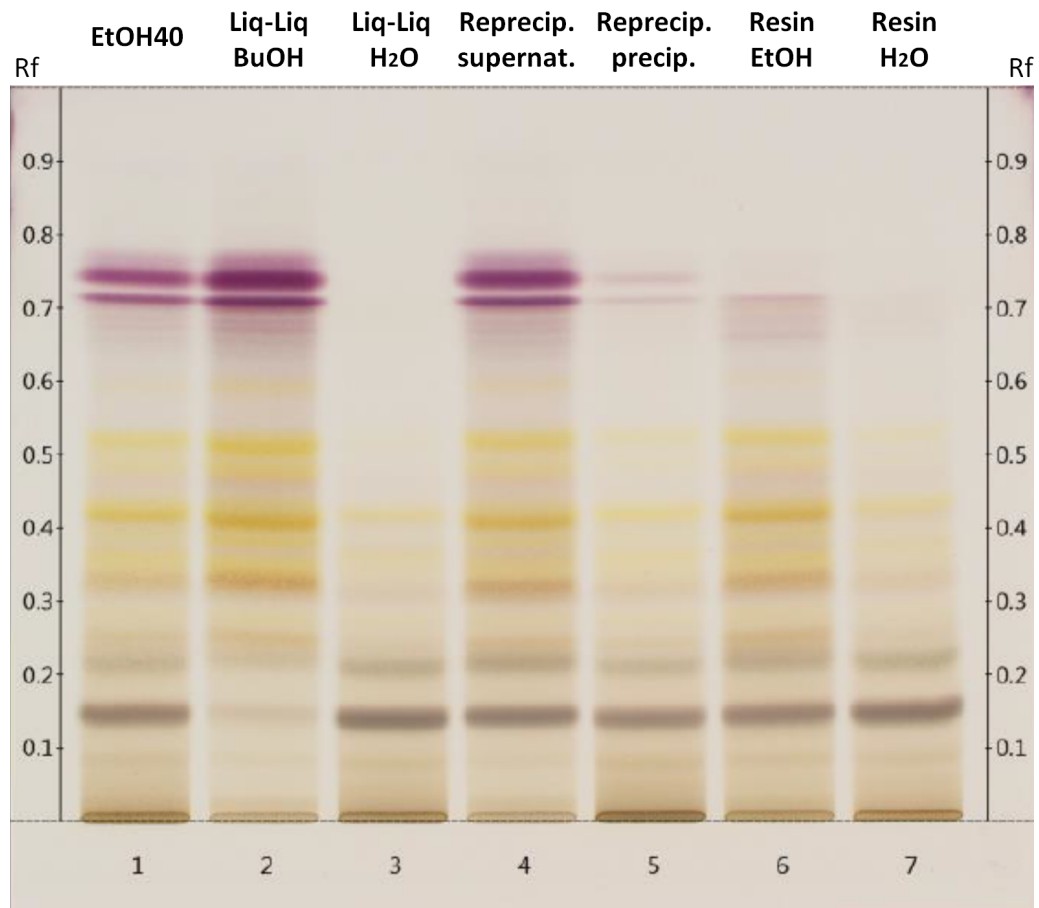

**Figure 3** **Thin layer chromatogram of polar constituents of 40% ethanolic *S. scardica* extract and its fractions after spraying with anisaldehyde solution R.** Allocation of the tracks: primary extract with 40% ethanol (1); butanolic (2) and aqueous (3) phase of liquid–liquid extraction; supernatant (4) and precipitated fraction (5) of reprecipitation in 70% ethanol; ethanolic (6) and aqueous (7) phase of resin separation.

control strain CL802 does not express Aβ. The progression of this paralysis was traced for at least 12 h (Fig. 5). The $PT_{50}$, a median value describing the point in time when exactly 50% of the worms were paralysed, was calculated to test for statistically significant differences. Extracts and fractions were tested in two sets of experiments, which showed slightly, but not significantly differing values of the negative control (0.5% ethanol) (Table 2). Nevertheless, all treatments were compared to the respective negative control of the test series. All worms treated with 600 µg/mL of the different *S. scardica* extracts showed a delay of the Aβ-induced paralysis similar to or better than the positive control 100 µg/mL EGCG. The most active extract was the one prepared from 50% ethanol (more than 10% delay), which was also acting in a concentration-dependent manner (Fig. 5B). 600 µg/mL of the 40% ethanolic extract showed about 5% delay, but was tested in another series of experiments, which makes a direct comparison inappropriate, especially as the percentage delay of EGCG also differs from 6 to 3% in the two sets. Treatment with 400 µg/mL

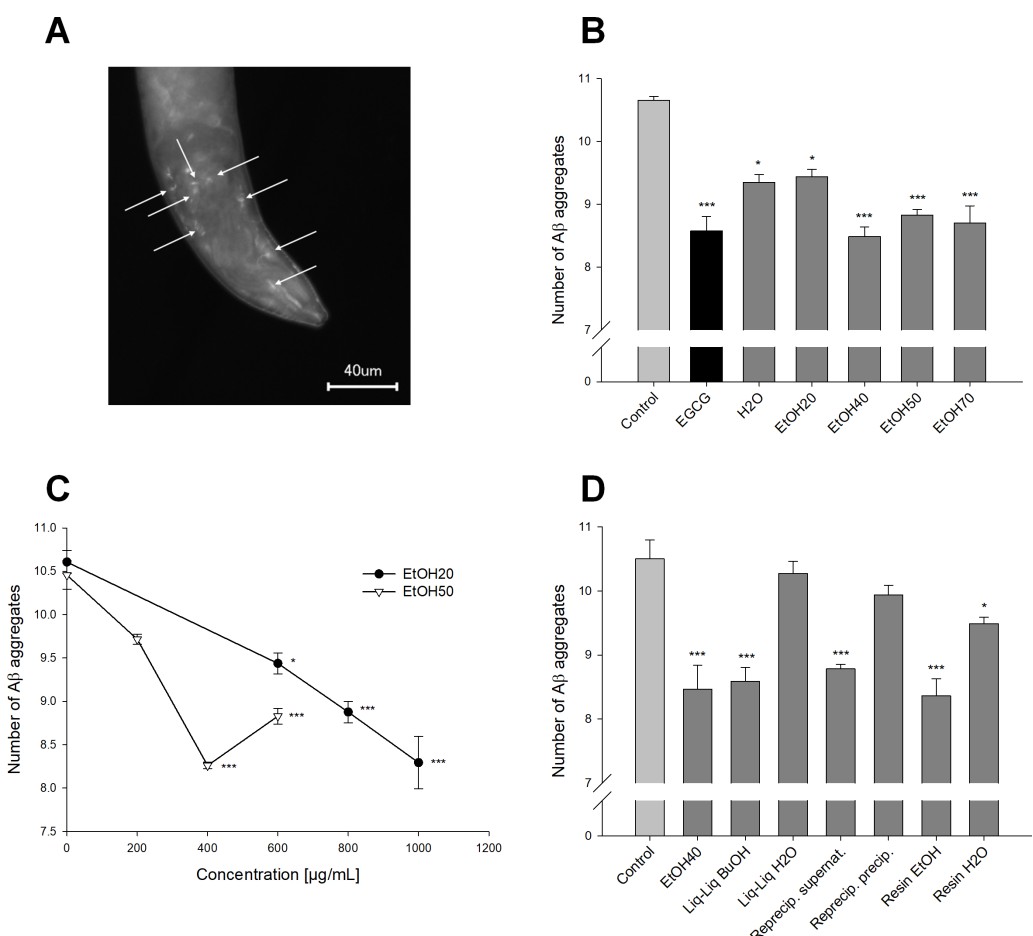

**Figure 4** **Effect of *Sideritis scardica* on Aβ aggregation in *C. elegans*.** (A) Fluorescence microscopic image of the head region of a worm from strain CL2006. Arrowheads point out the β-amyloid plaques that were stained with thioflavin S. (B, C, D) Reduction in number of the Aβ aggregates. All extracts were tested in a concentration of 600 μg/mL compared to 100 μg/mL EGCG as a positive control (B); the 20% and 50% ethanolic extract were tested in additional concentrations to show dose-dependence (C). All fractions were tested in a concentration of 400 μg/mL compared to the original 40% ethanolic extract in the same concentration (D). Controls were treated with 0.5% ethanol. $^*p < 0.05$; $^{***}p < 0.001$; concentrations in μg/mL.

of the more lipophilic extract fractions attenuated the progression of the Aβ-induced paralysis (about 6% delay each), whereas the polar fractions failed to increase the $PT_{50}$ significantly (Table 2; Fig. 5C). Worms of the control strain (not expressing Aβ) that were also treated with the extracts or fractions in the highest used concentration, did not exhibit any paralysis.

## DISCUSSION

All *S. scardica* extracts tested significantly reduced the number of Aβ aggregates and alleviated Aβ toxicity in transgenic *C. elegans* strains in a concentration of 600 μg/mL or less. Taken together, the 40 and 50% ethanolic extracts were the most active, although

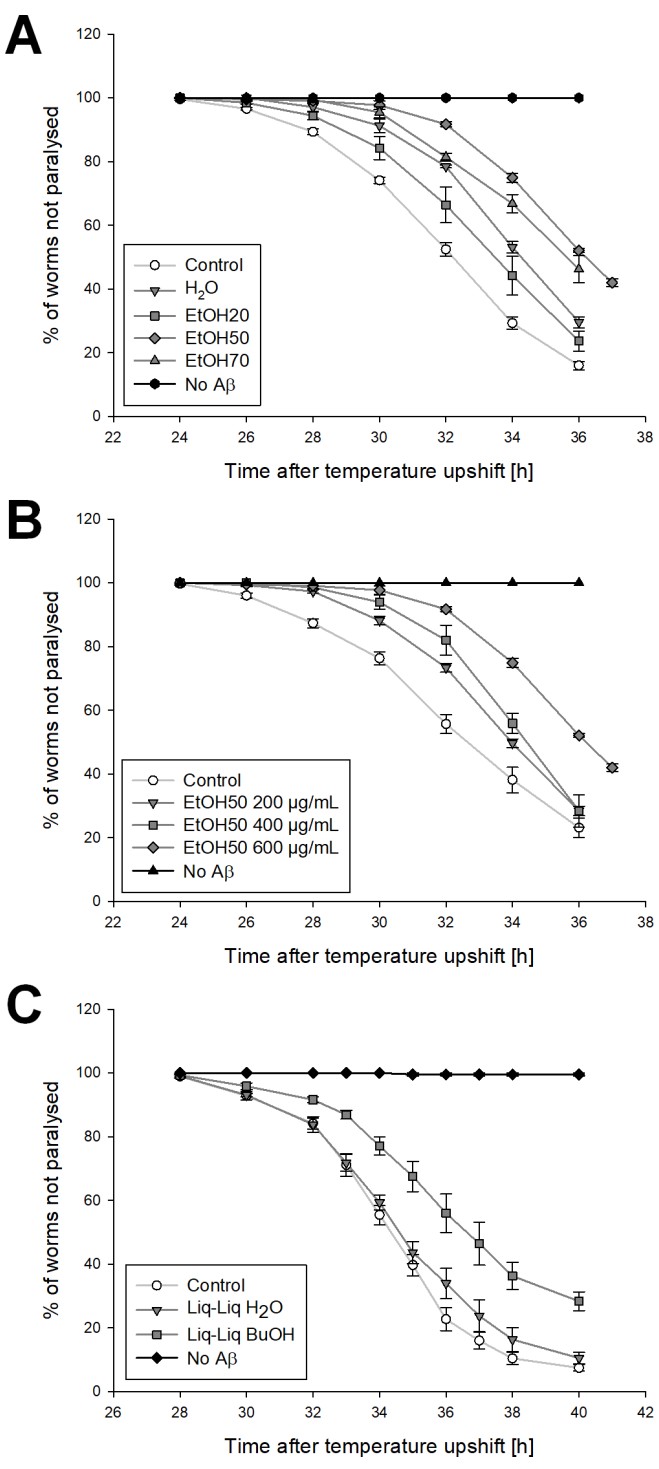

**Figure 5** **Influence of *S. scardica* on Aβ-induced paralysis.** (A, B) Paralysis curves from the first set of experiments. Compared to the control (0.5% ethanol) CL4176 worms treated with 600 µg/mL extract prepared from water, 20, 50, and 70% ethanol were paralysing slower and to a lesser extent (A). The control strain, which is not expressing Aβ, did not show any paralysis. Amongst

**Figure 5 (…continued)**
others, the 50% ethanolic extract showed a concentration-dependent activity (B). (C) Paralysis curves from the second set of experiments. All worms treated with 400 μg/mL of the more lipophilic extracts showed a delayed Aβ-induced paralysis, whereas 400 μg/mL of the polar ones failed to clearly shift the curve to the right. Results of the fractions from liquid–liquid extraction as an example.

**Table 2  Delay of Aβ-induced paralysis in *C. elegans* strain CL4176.** $PT_{50}$ are median values describing the point in time, when exactly 50% of the worms are paralysed. Experiments were conducted in different test series with slightly differing values of the control; $p$ values compared to the respective control.

| Series of experiments | Treatment | Concentration [μg/mL] | $PT_{50} \pm$ S.E.M. [h] | Significance |
|---|---|---|---|---|
| Set 1 | Control (0.5% ethanol) | | $33.5 \pm 0.5$ | |
| Set 1 | EGCG | 100 | $35.5 \pm 0.5$ | $p < 0.05$ |
| Set 1 | $H_2O$ | 600 | $35.5 \pm 0.5$ | $p < 0.05$ |
| Set 1 | EtOH20 | 600 | $35.0 \pm 0.6$ | $p < 0.05$ |
| | | 800 | $35.5 \pm 0.5$ | $p < 0.05$ |
| | | 1,000 | $35.8 \pm 0.6$ | $p < 0.05$ |
| Set 1 | EtOH50 | 200 | $35.0 \pm 0.6$ | |
| | | 400 | $35.5 \pm 0.5$ | $p < 0.05$ |
| | | 600 | $37.0 \pm 0.0$ | $p < 0.001$ |
| Set 1 | EtOH70 | 600 | $36.3 \pm 0.3$ | $p < 0.01$ |
| Set 2 | Control (0.5% ethanol) | | $34.8 \pm 0.3$ | |
| Set 2 | EGCG | 100 | $35.8 \pm 0.3$ | $p < 0.05$ |
| Set 2 | EtOH40 | 400 | $35.5 \pm 0.3$ | |
| | | 600 | $36.5 \pm 0.3$ | $p < 0.01$ |
| Set 2 | Liq-Liq BuOH | 400 | $37.0 \pm 0.6$ | $p < 0.05$ |
| Set 2 | Liq-Liq $H_2O$ | 400 | $35.3 \pm 0.3$ | |
| Set 2 | Reprecip. supernat. | 400 | $36.8 \pm 0.3$ | $p < 0.01$ |
| Set 2 | Reprecip. precip. | 400 | $35.5 \pm 0.3$ | |
| Set 2 | Resin EtOH | 400 | $37.0 \pm 0.4$ | $p < 0.01$ |
| Set 2 | Resin $H_2O$ | 400 | $35.3 \pm 0.3$ | |

EtOH40 did not show a similar percentage delay of paralysis. But a direct comparison of the values is difficult, as they were obtained from two different test series that were performed about 1.5 years apart. In an *in vivo* system like *C. elegans* a lot of factors, including behavior, can change to some extent. The worms of the second set basically started to paralyse some hours later and showed slightly different paralysis progression (see Fig. 5). But obviously data are consistent within the sets and none of the active substances tested in the same set showed a significantly higher activity than EtOH40. Thus, compared to extracts prepared from solvents of higher or lower polarity, the two mid-polar extracts showed the strongest activity, although they did not contain the highest content of flavonoids, acteoside, and caffeoylquinic acids as polyphenolic lead compounds. Reasons for this may be based on pharmacodynamic synergisms of certain extract constituents, as plant extracts always embody multicomponent mixtures. To elucidate this question, further studies must be performed. Also, bioavailability of active compounds or even bioenhancing effects may

play a role. Polyphenolics contained in *S. scardica* were reported to be bioavailable, as in a clinical study 5% of polyphenols ingested with a cup of tea were found as metabolites in urine samples via a HPLC-MS measurement (*Petreska & Stefova, 2013*). Drug absorption is not very well investigated in *C. elegans* itself, but doubtless compounds have to show similar properties as in vertebrates to be well absorbed from the intestine. According to *Zheng et al. (2013)* the amount of drugs being absorbed is similar in *C. elegans* and mice. Furthermore, the specifity of extraction solvents regarding the ratio of active to inactive constituents could lead to higher activity of the mid-polar extracts.

The 20 and 50% ethanolic extracts were tested in different concentrations, showing a dose-dependence in both assays. Only worms of strain CL2006 treated with 600 µg/mL EtOH50 did not show a lower number of Aβ plaques compared to 400 µg/mL. Beginning toxic effects at this concentration are highly improbable, as all chosen treatments were tested for their toxicity on *C. elegans* (data not shown). More likely, the extract is exhibiting a U-shaped dose–response curve, that is more realistic in biological systems than linear responses (*Calabrese & Baldwin, 2001*).

All the more lipophilic fractions of the primary 40% ethanolic extract showed a significant reduction in number of Aβ aggregates as well as a delayed Aβ-induced paralysis, with the level of activity being similar to the primary extract or just slightly higher, which points out that the lipophilicity of the extract constituents is perhaps not important alone, otherwise the extract prepared from 70% ethanol would have also shown better results than the mid-polar extracts. However, in most cases the polar fractions did not reveal significant effects, but show trends. So, synergistic effects, maybe of polar and unpolar constituents, are still worth being discussed and investigated. Considering the phytochemical profile of the extract fractions, it is not completely clear if their Aβ-counteracting activities can be attributed to their content of total polyphenols, or to a more specific class of compounds. But as the precipitated fraction of the reprecipitation of EtOH40 in 70% ethanol that contained more flavonoids than its lipophilic counterpart always showed lower activity, this group of compounds may not play a central role. Contemplating the enrichment of acteoside in the unpolar fractions, this phenylethanoid glycoside remains the most promising compound for a potential causal correlation of content and activity.

The Aβ-counteracting activity of hydroalcoholic *S. scardica* extracts has already been shown in mice (*Hofrichter et al., 2016*). Here the number of Aβ depositions, as well as the level of soluble Aβ (1–42) was decreased, which is coherent with the results of the present study. *Hofrichter et al. (2016)* also provided some evidence about the mode of action. They found an intensified Aβ clearance via enhancement of phagocytosis in microglia and induction of ADAM10 expression, a crucial $\alpha$-secretase, which cleaves Aβ (*Esch et al., 1990*). An induction of ABC transporter could not be found. An influence of *S. scardica* on secretases cannot be discussed using the results of the present study as the worms were expressing Aβ through a minigene, not by processing APP. Other possible mechanisms of action against Aβ toxicity involve anti-inflammatory and antioxidant activities (*Gilgun-Sherki, Melamed & Offen, 2001*; *Heneka et al., 2015*; *Kadowaki et al., 2005*; *Shelat et al., 2008*). As *S. scardica* has already shown anti-inflammatory properties (*Tadic et al., 2007*), the inhibition of neuroinflammation is a possible mechanism in vertebrates. But as the

nematodes are lacking important structures and mediators which promote inflammation, this is not applicable in the chosen model. Also, several antioxidant activities of Greek mountain tea have been described *in vitro* (*Todorova & Trendafilova, 2014*), but no antioxidant effects of the extracts or fractions, including the level of intracellular ROS (reactive oxygen species) and defense against the pro-oxidant compound juglone, was observed in *C. elegans* (data not shown). *S. scardica* is rich in polyphenols, which makes a direct interaction with Aβ peptides highly probable, as this is described for many polyphenolic compounds (*Porat, Abramowitz & Gazit, 2006*; *Stefani & Rigacci, 2013*). Assembly of peptides can be inhibited by hydrogen or ionic bonds (hydroxyl groups of polyphenols and amino groups of peptides), or by hydrophobic interactions. This direct inhibition of Aβ aggregation and oligomerisation is also described for EGCG, which was used as the positive control (*Abbas & Wink, 2010*; *Del Amo et al., 2012*; *Wang et al., 2010*). A reduced oligomerisation could likewise explain the alleviated Aβ toxicity.

## CONCLUSIONS

In conclusion, it can be stated that hydroethanolic *S. scardica* extracts inhibit Aβ aggregation and toxicity in *C. elegans* with the mid-polar extracts being the most active. This augments existing evidence and makes *S. scardica* highly interesting for the treatment or prevention of neurodegenerative diseases like Alzheimer's. Acteoside, a phenylethanoid glycoside, represents a promising, potentially active substance in the extracts and fractions. Nonetheless, further steps have to be taken to investigate the active principle of the extracts and potential synergistic actions of its constituents. In addition, a detailed mechanism of action cannot be stated at the moment; the hypothesized direct inhibition of Aβ aggregation needs further elucidation.

### Abbreviations

| | |
|---|---|
| **Aβ** | Amyloid-β |
| **ACN** | Acetonitrile |
| **AD** | Alzheimer's disease |
| **APP** | amyloid precursor protein |
| **DAD** | diode array detector |
| **DER** | drug-extract ratio |
| **EGCG** | (-)-epigallocatechin-3-gallate |
| **EtOH40** | 40% ethanolic *S. scardica* extract |
| **HMPC** | Committee on Herbal Medicinal Products |
| **MCI** | Mild cognitive impairment |
| **Rf** | Retardation factor |

## ACKNOWLEDGEMENTS

The *C. elegans* strains used in this study were provided by the CGC, which is funded by NIH Office of Research Infrastructure Programs (P40 OD010440). Special thanks go to Dr. Christopher Link, University of Colorado, for his great help concerning the strains he and his group created.

### Funding

The authors received no funding for this work.

### Competing Interests

Björn Feistel is head of scientific affairs at Finzelberg GmbH & Co. KG. Michael Wink is an Academic Editor for PeerJ.

### Author Contributions

- Felix Heiner conceived and designed the experiments, performed the experiments, analyzed the data, prepared figures and/or tables, approved the final draft.
- Björn Feistel conceived and designed the experiments, analyzed the data, contributed reagents/materials/analysis tools, authored or reviewed drafts of the paper, approved the final draft.
- Michael Wink analyzed the data, contributed reagents/materials/analysis tools, authored or reviewed drafts of the paper, approved the final draft.

### Supplemental Information

Supplemental information for this article can be found online at http://dx.doi.org/10.7717/peerj.4683#supplemental-information.

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
