# Peer review of "Sideritis scardica extracts inhibit aggregation and toxicity of amyloid-β in Caenorhabditis elegans used as a model for Alzheimer’s disease"

_PeerJ, doi:10.7717/peerj.4683_

## Round 0.1 · original submission · Major Revisions

Dear authors,

The manuscript "Sideritis scardica extracts inhibit aggregation and toxicity of amyloid-β in Caenorhabditis elegans used as a model for Alzheimer's disease " requires Major Revisions.

Minor corrections:

- In line 92 and in Figure 1, Sideritis scardica is bad written. Please, correct it.

Reviewer 1 ·

Basic reporting

no comment

Experimental design

The methods employed are generally described sufficiently and to an extent that allows for replication. However, a few minor questions came up at some points, which I think could improve the replicability:
• Could you please add a little bit more information on how the solid-liquid separation was performed? I.e., what solvent was used to dissolve the extract in the beginning? Were both solvents, ethanol and water used subsequently to elute the respective compounds from the resin?
• How did you identify the specific polyphenolic compounds by HPLC? E.g. how were flavonoids distinguished from the other polyphenols in the mixture? Please add this information very briefly or provide a reference, if the method has been established before.
• Thank you for providing the results of the thin layer chromatography! I think this is a very valuable method (however, often underestimated) to obtain fast and good overviews to easily compare the composition of extracts and fractions. The picture of the plate in figure 2 was given in grayscale; would it be possible to provide a coloured picture? This would enable the reader to better distinguish between the compound classes.
• Could you please add the amount of time the worms were treated in hypochlorite solution? As this is the standard procedure for age-synchronized cultures, I don’t think it has to be described in detail, but since the concentration of NaOCl and NaOH may vary among protocols, the duration of the treatment would be necessary for replication.
• Also, it would be helpful for the reader to add the information about the respective mutant C. elegans strains somewhere to the Materials and Methods section. All necessary information is given within the figures or results, but I would suggest to additionally present it at the beginning.
• In line 142, it reads “…to induce the expression”; probably the expression of the Aβ peptide? Please add this information.

Validity of the findings

no comment

Additional comments

Results and discussion are conclusive and cover all of the research questions presented in the beginning. The data presented are statistically sound with a sufficient number of replicates and individuals treated, negative and positive controls are included in every experiment. However, a few questions / remarks remain:
• As you stated in line 243/244, acteoside might play a major role for the bioactivity of the extracts. I think it would have been interesting to include the isolated compound (which I think is commercially available) in your bioassays to see whether it exerts an activity itself, and if yes, to what extent compared to the whole extracts. Why wasn’t the compound included?
• You mentioned the presence of tannins (line 168). As their Rf values are rather high (0.7 – 0.8) using the mobile phase described, the respective compounds are presumably rather small and therefore lipophilic. If this information is available, could you specify on the type of tannins? Also, EGCG is used as the positive control, which could be regarded as a monomeric unit of a condensed tannin, and you mentioned tannins to be present in the lipophilic / active fractions; couldn’t this compound class contribute to the bioactivity as well?
• Regarding figure 3: Is the example shown in picture A the 0.5 % EtOH control or a treated worm? Please indicate the treatment in the legend. An additional picture showing the 0.5 % EtOH control versus a treated worm (e.g. EtOH40) could visualize the effect more clearly.
• Figure 4: I suggest to specifiy “control” in the legend within the graph a bit more in detail (e.g. “control EtOH 0.5 %) and to indicate that “No Aβ” means that a different strain was used (e.g. “No Aβ (strain CL…)”). Although the information is given in the legend, reading only the figures, “control” could be confused with EGCG.
• Line 189/190: I suggest to write “negative control”, instead of just “control”, because you previously mention the “positive control EGCG” for the plaque reduction assay and the “control strain CL802”, so the term “control” is not immediately clear from the text.
• Line 168: “Rf 0,3… “: please replace “,” by “.”

Reviewer 2 ·

Basic reporting

1. Please include relevant information for strains NL5901 and CL802 that would make them appropriate controls respectively for the aggregation formation and paralysis experiments, respectively.

2. Please include the deviation in the percent (%) reduction of aggregates observed per treatment group. For example, there was 21% ± (deviation)%, and not the mean alone.

3. Even though the experimental design involved using 20, 40, 50, and 70% ethanol extractions, and used 200, 400, 600, 800, and up to 1000 ug/mL concentrations for treatment, there appears to be only a select number of data presented in the paper. If certain data were presented, while others were not presented, can you please provide a rationale for this decision?

Experimental design

1. The C. elegans strains used (CL2006 & CL4176) express amyloid-beta in muscle cells, and are therefore expressing these aggregatory proteins in the whole body. Is there any reason why only the head region was observed?

2. Why was the analysis limited to the number of amyloid-beta aggregates, and the size of amyloid-beta aggregates ignored? Were all the amyloid-beta aggregates formed (and observed) the same size? Instead of counting aggregates, would it be possible to quantify the total amount of amyloid-beta aggregates by summing up the fluorescence of the aggregates?

3. Some researches have mentioned that for strain CL4176, there is already a considerable expression of amyloid-beta and formation of aggregates even at 16 degrees Celsius before the upshift in temperature. In your hands, did you also observe this?

4. Is there a reason why there were no data points plotted for EtOH50 beyond 600 ug/mL in Figure 3-C?

Validity of the findings

1. Can you give an explanation as to why the total phenols was ‘surprisingly high’ in the 40% ethanol fraction?

Additional comments

No comment.

---

## Round 0.2 · accepted · Accept

Dear authors,

Your manuscript has been accepted for publication in PeerJ.

# Reviewer 1 ·

Basic reporting

no comment

Experimental design

no comment

Validity of the findings

no comment